# Mechanical Properties of Carbon-Fiber RPC and Design Method of Carbon-Fiber Content under Different Curing Systems

**DOI:** 10.3390/ma12223759

**Published:** 2019-11-15

**Authors:** Xuejian Zhang, Lincai Ge, Yunlong Zhang, Jing Wang

**Affiliations:** 1School of Materials Science and Engineering, Jilin Jianzhu University, Changchun 130000, China; zxj_0620@163.com (X.Z.); gelincai@outlook.com (L.G.); 2School of Transportation Science and Engineering, Jilin Jianzhu University, Changchun 130000, China

**Keywords:** building materials, reactive powder concrete, carbon-fiber content, curing system, mechanical properties, microscopic, calculation formula

## Abstract

Natural, standard, and compound curing are adopted to study the effect of different curing systems on the reinforcement of carbon fiber in reactive powder concrete (RPC). This work systematically studies the changes in RPC compressive and tensile strengths under different curing systems. Taking age, fiber content, and curing system as parameters, Scanning electron microscope (SEM) and X-ray diffraction (XRD) microscopic methods are used to study the influencing mechanism of carbon-fiber content and curing systems on RPC. The calculation methods of the RPC strength of different carbon-fiber contents are studied. Results show that the optimum carbon-fiber content of carbon-fiber RPC is 0.75% under the natural, standard, and compound curing conditions. In comparison with standard curing, compound curing can improve the early strength of carbon-fiber RPC and slightly affect the improvement of late strength. The strength is slightly lower in natural curing than in standard curing, but the former basically meets the requirements of the project and is beneficial for the practical application of this project. The calculation formula of 28-day compressive and splitting tensile strengths of carbon-fiber content from 0% to 0.75% is proposed to select the carbon-fiber content flexibly to satisfy different engineering requirements.

## 1. Introduction

Reactive powder concrete (RPC) is a new type of cement-based composite material, which was first developed in Bouygues, France in 1993. This material is a cementitious composite with Portland cement and various mineral admixtures [1,2]. The basic principle of RPC formulation is to increase the activity and fineness of each part of the material. The content and amount of ultrafine powders, such as silica fume, are used to achieve a low water/binder ratio (W/B) for diminishing the porosity and obtaining optimal strength [3]. Such concrete exhibits several advantages, such as light weight, high strength, high toughness, and high durability. RPC is applied in various industries, including petroleum, nuclear power, municipal, marine, and military facilities, and it achieves high social benefits [4].

Numerous scholars worldwide conducted considerable research on steel-fiber RPC. The tests confirmed that steel fiber can significantly improve the mechanical properties of RPC [5,6,7,8]. However, steel-fiber corrosion remains a problem. Carbon fiber is a microcrystalline graphite material. Such a material has properties similar to carbon, including a stable structure, high-temperature resistance, high tensile strength, high modulus of elasticity, corrosion resistance, small specific gravity, and strong toughness. Moreover, carbon fiber is frequently used to increase, strengthen, cure, and reinforce concrete materials [9,10,11]. Khayat et al. studied the influence of carbon nanofibers on the mechanical properties of ultra-high-performance concrete; these authors concluded that the carbon-nanofiber content gradually increased from 0% to 0.3%, and its tensile and flexural strengths increased by 56% and 59%, respectively [12]. The addition of carbon fiber can significantly increase the compressive strength and reduce the electrical resistivity of the cement matrix [13]. Rangelov et al. studied the different sizes of carbon fibers and incorporated them into cement mortar mixtures. The 28-day compressive strength increased by up to 11%, and the seven-day tensile strength increased by up to 56% [14]. Mastali et al. studied the effects of carbon fiber after adding 0.5%, 1%, 1.5%, and 2% with lengths of 10, 20, and 30 mm on the mechanical properties of cement mortar. The mechanical properties and impact resistance of the composites significantly improved with the increase in volume fraction and length, but the construction performance was reduced [15]. Hossain utilized other carbon fibers with lengths of 3, 6, and (3 + 6) mm with 0%, 0.5%, 1%, and 1.5%. The load–deflection curve was used for quantitative analysis. The optimal effect was found for (3 + 6) mm, and the optimal dosage was 1.5% [16]. Previous research on carbon-fiber-reinforced cement-based materials mainly focused on cement mortar containing coarse aggregate and a high water-to-binder ratio. Carbon-fiber-reinforced RPCs with low water-to-binder ratios are rarely reported. This phenomenon is due to the difficulty in dispersing carbon fiber and its easy agglomeration. Therefore, the present work tests the compressive and tensile strengths of RPC with different fiber contents to meet actual engineering requirements, and it provides a calculation formula of the carbon-fiber content.

Fiber-reinforced RPC composites, in addition to the type, size, and amount of fibers, positively or negatively affect the cement matrix. Existing research suggested that the curing method also significantly impacts the micromorphology and mechanical properties of RPC. However, no standard RPC curing condition is available at present. Under low-temperature curing conditions, the strength of concrete decreases, and the curing temperature is crucial for the strength improvement of cement-based materials [17,18]. Generally, the RPC preparation typically adopts two kinds of curing methods: the first one is heat curing, and the other one is autoclave curing; the former includes steam, hot water, and high-temperature curing conditions [19]. In heat curing, the temperature can be different, and the methods can promote the RPC hydration process [20]; the microstructure and crystal type of the cement matrix change accordingly. The C–S–H chain length increases at 90 °C, and no special crystals are found at 100 °C. Calcite is produced at 150 °C and develops at 200 °C. Hard xonotlite is produced at 250 °C [21,22]. Autoclave curing can reach the normal 28-day standard curing state in a short time. An increase in temperature and pressure changes the chemistry of the hydration product. Simultaneously, the total pore volume is reduced given the pressure difference, but the capillary pore volume is increased to provide the required space for the growth of C–S–H [23]. Autoclave and heat curing cannot be completed at the construction site. Such processes can only be accomplished in the prefabrication field and laboratory. This feature is non-conducive to the promotion and application of RPC. Therefore, RPC must be studied under the effects of natural, standard, and compound curing to provide a theoretical foundation for its application in practical engineering.

The mineral admixture in this test was a silica fume–mineral powder–fly ash cementing material system. Through indoor experiments, the compressive and tensile strengths of carbon fiber blended with different amounts of carbon-fiber RPC at various ages were investigated through natural, standard, and combined curing methods. The microscopic mechanism was analyzed by SEM and X-ray diffraction (XRD). The experimental results demonstrated the effects of carbon-fiber volume and curing age on the mechanical properties of carbon-fiber RPC.

## 2. Materials and Methods

### 2.1. Test Materials


(1)Cement: Yatai P.II 52.5 grade cement with a specific surface area of 367 m^2^/kg was selected. Table 1 shows the chemical composition and physical properties of the cement.(2)Silica fume: The micro-silica ash produced by Shandong Boken Silicon Material Co., Ltd. (Dongyue, China) was used, and the activity index was 116%. Table 2 summarizes the silica fume composition.(3)Fly ash: Secondary fly ash was obtained from a power plant in Jilin.(4)Slag-powder: An ordinary S95 slag-powder was acquired from a company in Gongyi, China. Table 3 lists the performance indicators.(5)Water-reducing agent: An HSC polycarboxylic acid high-performance water-reducing agent produced by Qingdao Hongxia Polymer Material Co., Ltd. (Qingdao, China) was used, and the water-reducing rate was 28.5%.(6)Quartz sand: The selected quartz sand was natural river sand. Such sand was obtained in Jilin Province. The sand with particle size below 1.18 mm was selected as the test sand. The labor cost of such sand was reduced in comparison with that of the sieved fine sand. The fineness modulus was 2.158, which is a fine sand value.(7)Carbon fiber: The carbon fiber produced by a company in Shandong was used. Table 4 lists the performance indexes.


### 2.2. Preparation and Curing of Test Pieces

The test used 100 mm × 100 mm × 100 mm concrete compressive strength standard test pieces. The production and vibration of the test piece were carried out in accordance with the specifications in “RPC” (GB/T 31387-2015). After completing the manual vibration, the test pieces were placed on the vibrating table and vibrated until no obvious bubble overflows were observed. The time was between 45 and 75 s. After forming the test pieces, they were placed in a standard curing box with a temperature of 20 °C and humidity of 95%. After 24 h, the test pieces were removed and numbered. The processes are described in Table 5.

The stirring process is shown in the Figure 1, the curing mode is shown in Figure 2, and the experimental mix is shown in Table 6.

## 3. Test Results and Discussion

### 3.1. The Working Performance of Carbon-Fiber RPC

The working performance of RPC with different carbon-fiber content is shown in Table 7.

As it can be seen from Table 7, carbon-fiber content had a great influence on RPC liquidity. When the content was 0.5%, the fluidity of carbon-fiber RPC decreased by 15 mm. When the content was 0.75, the fluidity of carbon-fiber RPC was 210 mm, and, when the content was 1.0%, the fluidity dropped to 175 mm, and the mixing material of carbon-fiber RPC became more viscous. When the fiber content was 1.25%, the fluidity was 155 mm, the specimen was difficult to form, and the working performance became poor. According to the above analysis, considering RPC workability, when the fiber content was 0.5% and 0.75%, there was good fluidity, which is recommended for engineering applications.

### 3.2. Compressive and Splitting Tensile Strength Test Results

The test results of compression strength and splitting strength of carbon-fiber RPC under natural curing, standard curing, and composite curing on days seven and 28 are summarized in Table 8.

### 3.3. Analysis of the Pressure Resistance and Splitting Tensile Strength of the Curing System

#### 3.3.1. Analysis of the Results of Cubic Compressive Strength Test and Failure Modes

Figure 3a exhibits that, when the concrete was not mixed with carbon fiber, multiple cracks slowly appeared in the concrete during the process of load increase. When the peak stress was reached, the cracks gradually expanded and extended and interlocked, showing the phenomenon of skin peeling through the cracks. As the load continued to increase, the spalling phenomenon became more and more serious, and then brittle failure directly occurred after a loud noise. Figure 3b demonstrates that, after adding carbon fiber, cracks gradually appeared in concrete along with the continuous increase of load. The carbon fiber and cement matrix formed a good constraint grid to limit the development of cracks and prevent the generation of through cracks. A loud noise was also generated when the ultimate load damage was reached. However, the integrity was better in the test piece than in the unblended fiber. The test phenomenon showed that the carbon-fiber inclusion could significantly alleviate the brittle characteristics of RPC, and the degree of brittle failure could be significantly reduced. Therefore, the integrity of the test piece was improved.

Figure 4 shows the compressive strength of seven-day and 28-day carbon-fiber RPC under the natural, standard, and compound curing conditions. The effect of the three curing conditions on seven-day compressive strength was in the order compound curing > standard curing > natural curing. By contrast, the effect on 28-day compressive strength was in the order standard curing > compound curing > natural curing. Under the natural and standard curing conditions, the compressive strength curves of seven and 28 days were relatively consistent, and the optimal blending amount was achieved when the carbon-fiber content was 0.75%. The compressive pressure was obtained when the fiber content exceeded 0.75. The intensity gradually decreased. Under the standard curing conditions, the external free water of the carbon-fiber RPC entered the cement matrix through the capillary pores to promote the hydration process. This situation was due to the obvious differences in temperature and humidity between the two curing methods. In the compound curing mode, the compressive strength of RPC at seven days was lower than that at 28 days when the fiber was not blended. The compressive strength at 28 days was significantly higher than that of uncarbonized fiber RPC with the increase in the carbon-fiber content. The reactants continuously filled between the pores and the cracks with the progress of the hydration reaction. At 28 days, the micro-cracks in the carbon-fiber RPC structure were reduced. This result was beneficial for the increase in compressive strength.

Figure 5 exhibits that, when the carbon-fiber content did not reach the agglomeration limit, the fiber addition effectively improved the RPC compressive strength because the carbon fiber fully exerted a crack-blocking effect and improved the weak interface in the cement collection. At seven days, carbon-fiber RPC with fiber content of 0.5%, 0.75%, 1.0%, and 1.25% was compared with the RPC without fiber. The results at seven days show that the compressive strength of carbon-fiber RPC under natural curing condition increased by 7%, 13.4%, 2.7%, and 4.29%, respectively. Furthermore, the compressive strength of carbon-fiber RPC under standard curing condition increased by 5.3%, 19.4%, 0.38%, and −2.6%, respectively. Under natural curing and standard curing conditions, when the carbon-fiber content was greater than the critical value of 0.75%, fiber agglomeration inside the RPC matrix occurred, thereby resulting in an increase in the internal defects and a decrease in compressive strength. At 28 days, carbon-fiber RPC with fiber content of 0.5%, 0.75%, 1.0%, and 1.25% was compared with the RPC without fiber. The results at 28 days show that the compressive strength of carbon-fiber RPC under natural curing condition increased by 11.5%, 28%, 19.9%, and 18.9%, respectively. Furthermore, the compressive strength of carbon-fiber RPC under standard curing condition increased by 12.04%, 23.1%, 14.2%, and 2.7%, respectively. Compared with plain RPC, the compressive strength of carbon-fiber RPC increased by −1.7%, 8.3%, 6.4%, and 1.4% in the compound curing condition. After adding the carbon fiber, the fiber exhibited a three-dimensional (3D) disordered distribution in the cement matrix, which had a certain inhibitory effect on the initial crack development. Carbon fiber was dispersed in the cement matrix and had a bridging effect, thereby positively promoting the compressive and splitting tensile strength of RPC.

#### 3.3.2. Analysis of the Results of Cubic Splitting Tensile Strength Test and Failure Modes

Figure 6a,b demonstrate that the specimen was brittle and even directly flew out when the prime RPC was tested in the pull-up test, and the ultimate load was reached. When the carbon fiber was incorporated into the RPC matrix, micro-cracks were generated during the loading process, extending from the upper and lower sides. Subsequently, cracks were formed on the sides. When the ultimate load was reached, the upper and lower cracks directly penetrated the sample, and the test block was indirectly separated. The carbon-fiber RPC’s splitting tensile strength significantly increased in comparison with the RPC. 

Figure 7a,b exhibit the splitting tensile strength of carbon-fiber RPC under natural, standard, and compound curing conditions at seven and 28 days. The effect of the three curing conditions on the seven-day splitting tensile strength was in the order compound curing > standard curing > natural curing. By contrast, the effect on the 28-day splitting tensile strength was in the order standard curing > compound curing > natural curing. When the fiber was not blended, the splitting tensile strength of RPC at seven days was lower than that at 28 days. For the natural compressive strength and standard curing mode, the seven-day and 28-day compressive and splitting tensile strengths initially increased and then decreased with the increase in carbon-fiber content.

Figure 8 illustrates that, when the fiber content was 0.5%, 0.75%, 1.0%, and 1.25% for the carbon-fiber concrete at seven days, its splitting tensile strength under standard curing condition increased by 17.7%, 58.4%, 44.4%, and 60.7%, respectively, and increased by 17.7%, 58.4%, 44.4%, and 60.7% under standard curing condition compared with plain RPC. The strength of carbon-fiber RPC decreased under compound curing condition. The carbon-fiber RPC with the same content at 28 days showed an increase in splitting tensile strength by 41.6%, 67.3%, 38.2%, and 30.47% under natural curing condition, while it increased by 40.1%, 78.5%, 56.2%, and 62.7%, respectively, under standard curing condition. The seven-day strength was lower than that of plain RPC. This condition was due to the temperature difference between hot-water curing and natural curing during compound curing, which causes the capillary pores to create a high-temperature and high-pressure environment. An increase in the high temperature of the capillary void led to an increase in the degree of polymerization of the silicate gel, an apparent density increase of 25%, and a significant decrease in the bound water content. The microstructure of the cement matrix became rough and porous [24,25].

The effects of three curing methods on the mechanical properties of carbon-fiber RPC were based on standard curing. Figure 9a,b exhibit the seven-day and 28-day compressive and splitting tensile strength ratio and the compound and standard curing compressive and splitting tensile strength under natural and standard curing. For the ratio of strength at seven and 28 days under natural curing conditions, the carbon-fiber RPC splitting tensile strength reached 68.7% of the value for standard curing, and the compressive strength reached more than 90% of the value for standard curing. At seven days under compound curing conditions, the compressive and splitting tensile strengths of carbon-fiber RPC accounted for >90% and 68% of the values for standard curing. When the carbon-fiber content was 0.5%, the compressive and tensile strengths accounted for 127.26% and 156.7% of the standard curing, respectively. At 28 days, the compressive strength under compound curing accounted for >100% of the value for standard curing, and the splitting tensile strength exceeded 75% of the value. This increment was due to the high proportion of carbon-fiber RPC gelling components. The development of hydration products was promoted, and the volcanic ash reaction between silica fume and Ca(OH)_2_ was accelerated, thereby resulting in a seven-day composite. These occurrences were attributed to the high percentage of hydration products produced by the rapid hydration of compound curing in the previous one-day hot-water curing conditions. The intensity of curing was high. With the increase in age, compound curing was in the natural curing stage, the free water and the combined water supply for the hydration reaction were insufficient, and the hydration process was slow. Accordingly, the intensity of standard curing gradually decreased at 28 days.

## 4. RPC Micromechanics Research

### 4.1. SEM Microanalysis

An SEM microstructural analysis is carried out to study the carbon-fiber distribution in the cement matrix and the bonding properties of fiber and interface. With the increase in carbon-fiber content, the carbon-fiber RPC’s mechanical properties, crack resistance, and interface strength were improved. Figure 10a,b exhibit that carbon fiber and cement had good compatibility and bond closely. As discussed in Section 3.3, when the fiber content changed from 0–0.75%, the compressive strength and splitting tensile strength of carbon-fiber RPC increased with the increase in carbon-fiber content. The fiber and the matrix had a good bonding effect, and the carbon fiber bore part of the external load. When the fiber and the matrix cooperated, most of the fibers were pulled apart rather than pulled out. When the carbon-fiber content was 0.75%, more fibers were pulled off, and more energy was absorbed; thus, the compression and tensile strength were strengthened. From this point of view, the proper amount of carbon fiber can improve the mechanical properties of RPC. As shown in Figure 10c,d, when the fiber content was 1%, internal defects occurred, such as agglomeration and overlap of the carbon fiber. The compressive strength and splitting tensile strength of RPC were higher when the content of carbon fiber was 0.75% rather than 1%. When the fiber content was 1.25%, the fiber agglomeration and overlap were serious, and numerous defect regions were formed inside the cement matrix. This condition further reduced the mechanical properties of RPC. Therefore, as mentioned in Section 3.3, when the carbon-fiber content exceeded the critical value of 0.75%, the fiber agglomeration phenomenon inside the RPC matrix was serious, leading to increased internal defects and decreased compressive strength.

### 4.2. XRD Microanalysis

In order to analyze the hydration mechanism of carbon-fiber RPC at different ages under different curing systems, XRD patterns of RPC at seven days and 28 days were studied under different curing conditions. Figure 11a,b depict the XRD patterns of RPC at seven and 28 days using different curing systems. Given that RPC is a cement-based composite with a low water-to-binder ratio, numerous minerals are unhydrated. The intensity of the diffraction peaks of Ca(OH)_2_ and C–S–H at seven days was greater under compound curing than under standard and natural curing. This situation was attributed to silica fume–mineral powder–fly ash with a high activity of silica fume in the cementitious material system with a “volcanic ash effect” under the one-day 90 °C hot-water curing condition. The thermal excitation effect was evident. The reactivity was higher in the slag-powder than in the fly ash. During hot-water curing, the silica ash and ore powder are involved in secondary hydration to increase the number of hydration products. The porosity is reduced, and the structure is compact. Under the standard and natural curing conditions, RPC hydration relies on the internal crystallization water. By contrast, the considerable mineral admixture has a long secondary hydration reaction at normal temperature and a low degree of hydration. At seven days, a strong peak of silicate stone appeared at 2θ = 27.8° and 29.5° under compound curing. This situation also explains the product of autoclave curing of RPC material under heat curing. The structure was dense, given the presence of calcium silicate crystals. These crystals were the root cause of the high strength of RPC materials at seven days under compound curing. At 28 days, the form of calcium silicate crystals changed, and the pores increased. This phenomenon was also the reason for the strength growth in the late stage of compound curing slowing or even decreasing. The diffraction peak of Ca(OH)_2_ was weak. The high temperature, accelerating the hydration in the initial stage, led to the premature formation of a thick C–S–H gel layer on the surface of the cement particles. This occurrence affected the later hydration [26], but the Ca(OH)_2_ diffraction peaks under standard and natural curing were significantly higher than those under compound curing. Accordingly, a good alkaline environment was created for the secondary rehydration of RPC. Therefore, the strength growth was better in RPC under compound curing than under standard and natural curing conditions.

## 5. Carbon-Fiber Content Design

### 5.1. Carbon-Fiber RPC Compressive Strength Design

This work aimed to determine the relationship between carbon-fiber content and compressive strength. The average of the percentage increase in the compressive strength of RPC at 28 days was fitted under the three curing modes on the basis of the compressive strength increase of RPC at 28 days. Figure 4b displays that, when the fiber content exceeded 1%, the strength increase rate started decreasing, and the preparation cost became high. This work selected 0.75% as the demarcation point. The recommended carbon-fiber amount in the actual project should be 0–0.75%. The high amount of carbon-fiber agglomeration is serious and is likely to decrease the strength with the increase in fiber content. The present work adopted linear fitting to facilitate the engineering needs (Figure 5b). The formula for calculating the compressive strength was obtained in accordance with the percentage increase in carbon-fiber RPC compressive strength at 28 days.
(1)$$f=\left(1+\lambda_{c}\right)f_{0}$$
where *f* is the 28-day compressive strength (MPa), $\lambda_{c}$ is the carbon-fiber compressive strength contribution, $\lambda_{c}$ = $\frac{25x-1.1}{100}$ where *x* is the carbon-fiber content (%), and *f*_0_ corresponds to the 28-day compressive strength of carbon fiber without the curing mode (MPa).

The experimental results of this work were compared with the calculated ones to verify the practicality of the carbon-fiber RPC compressive strength formula (Table 9). Figure 12 plots the calculated and test values under natural, standard, and compound curing conditions.

Ke [27] and Guo [28] investigated the effects of carbon-fiber content on RPC strength under different curing conditions; these studies aimed to further verify the relationship’s feasibility with regard to carbon-fiber content and compressive strength. The calculated values of this formula were compared with the test results of the aforementioned authors (Table 10).

Table 10 illustrates that the error between the calculated and actual values was ≤7%. This finding agrees well with the test results. This formula can be applied to carbon-fiber RPC when calculating the 28-day compressive strength of carbon-fiber RPC under natural, standard, and hot-water and steam curing conditions from 0% to 0.75%.

### 5.2. Carbon-Fiber RPC Splitting Tensile Strength Design

The percentage of carbon-fiber blending under the three curing modes was averaged for the increase in splitting tensile strength of RPC at 28 days to design the relationship between the carbon-fiber content and the splitting tensile strength. Linear fitting was performed using a carbon-fiber content of 0–0.75% for the average increase in splitting tensile strength. Figure 8b demonstrates that the equation for calculating the splitting tensile strength of carbon-fiber RPC at 28 days can be obtained using the formula of the percentage increase in splitting tensile strength.
(2)$$f=\left(1+\lambda_{t}\right)f_{0},$$
where *f* is the 28-day splitting tensile strength (MPa), $\lambda_{t}$ is the carbon-fiber splitting tensile strength contribution, $\lambda_{t}$ = $\frac{91x+1.25}{100}$ where *x* is the carbon-fiber content (%), and *f*_0_ is the 28-day splitting tensile strength of uncarbonized fiber under the corresponding curing mode (MPa).

In order to verify the applicability of the carbon-fiber RPC splitting tensile strength formula, Because the splitting tensile strength of RPC was significantly affected by the size and type of carbon fiber, verification through the use of other research conclusions was infeasible. Experimental data were used to verify and compare the calculated values with the experimental ones (Table 11). Figure 13 exhibits the calculated and experimental values under the natural, standard, and compound curing conditions.

Table 11 displays that the error between the calculated and the actual values was within 10%. The degree of compliance with the test results is good. This formula is suitable for calculating the 28-day splitting tensile strength of carbon-fiber RPC under natural, standard, and compound curing modes when the carbon-fiber content is between 0% and 0.75%.

## 6. Conclusions

In this work, carbon-fiber RPC was prepared using chopped carbon fiber as a reinforcing material under natural, standard, and compound curing conditions. The design method of the carbon-fiber RPC mechanical properties and content was studied on the basis of the compressive and splitting tensile strength tests. The following conclusions were drawn:(1)The strength test results of different curing systems showed the order compound curing > standard curing > natural curing under the same mix ratio and process conditions. Under compound curing condition, the compressive strength and splitting tensile strength at seven days of carbon-fiber RPC were reduced to some extent. At 28 days, the compressive strength and splitting tensile strength began increasing with the advance of hydration. The natural curing strength was slightly lower than that of standard curing. However, the former basically meets the requirements of the project and is beneficial for practical application. In terms of the three curing systems, compound curing was better than standard and natural curing in terms of the 28-day compressive strength with the extension of age. In terms of the 28-day tensile strength, standard curing was better than compound and natural curing.(2)When the carbon-fiber content was between 0% and 0.75%, the compressive and splitting tensile strengths of carbon-fiber RPC were significantly improved with the increase in carbon-fiber content. The SEM and XRD microscopic analyses found that the carbon-fiber RPC porosity was extremely low, and the hydration products were mostly C–S–H gel. Compound curing can also produce the product of autoclave curing, namely, calcium silicate stone. The interfacial transition zone between the carbon fiber and the matrix was close. When the carbon fiber content was 0.75%, the probability of fiber agglomeration significantly increased, thereby adversely affecting the RPC strength.(3)The prediction formulas of 28-day compressive and splitting tensile strengths suitable for carbon-fiber content from 0% to 0.75% were proposed in accordance with the compressive and tensile strengths of carbon-fiber RPC at 28 days. The carbon-fiber content can be flexibly selected to satisfy different engineering requirements.

## Figures and Tables

**Figure 1 materials-12-03759-f001:**
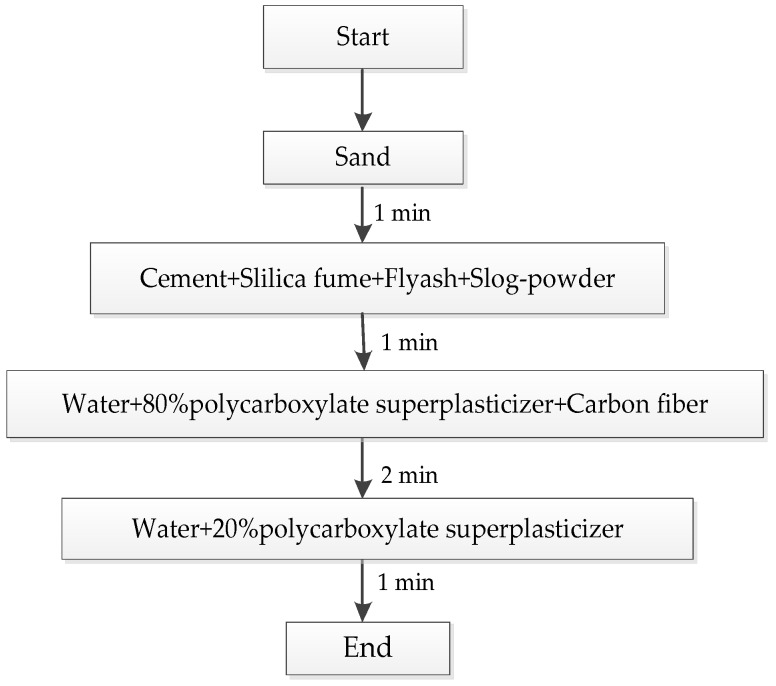
Stirring process.

**Figure 2 materials-12-03759-f002:**
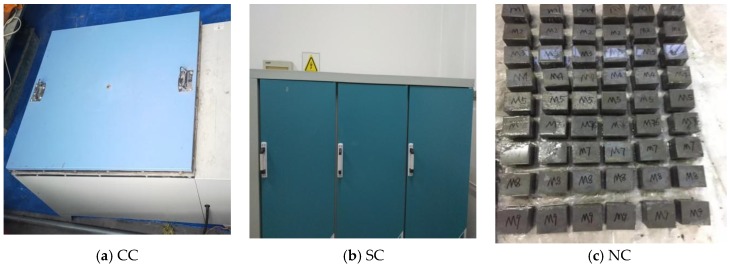
Curing methods: (**a**) hot-water curing, (**b**) standard curing, (**c**) natural curing.

**Figure 3 materials-12-03759-f003:**
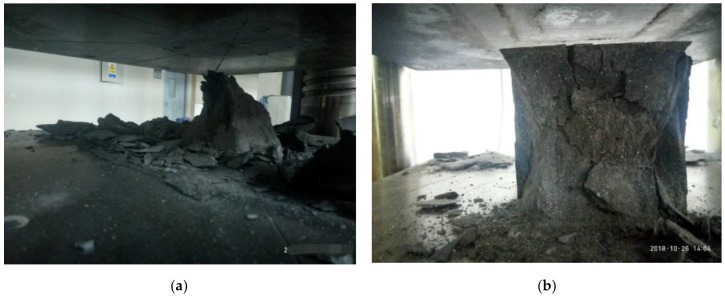
Compression failure morphology: (**a**) reactive powder concrete (RPC), (**b**) carbon-fiber RPC.

**Figure 4 materials-12-03759-f004:**
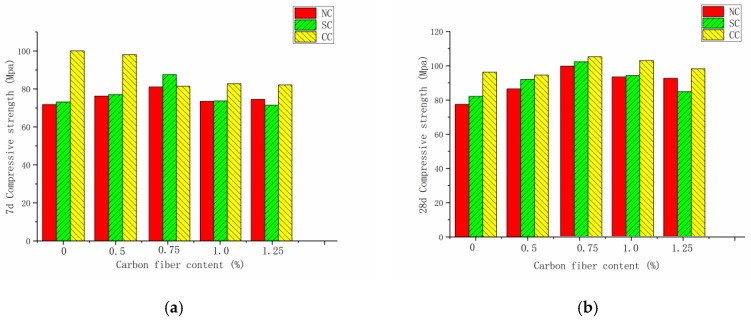
Compressive strength under different curing systems at seven days and 28 days: (**a**) seven days, (**b**) 28 days.

**Figure 5 materials-12-03759-f005:**
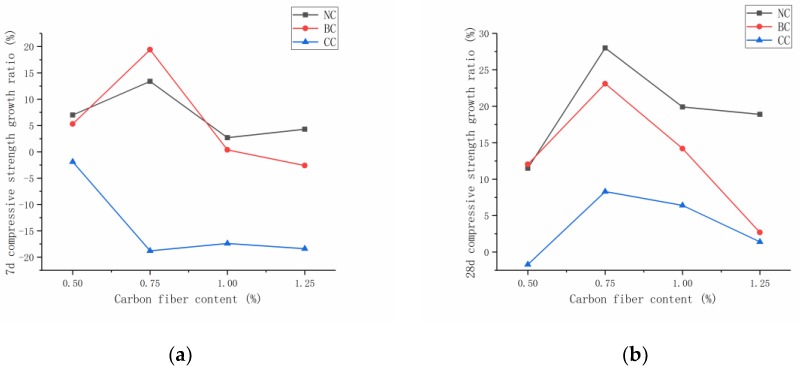
Compressive strength growth ratio at seven days and 28 days: (**a**) seven days, (**b**) 28 days.

**Figure 6 materials-12-03759-f006:**
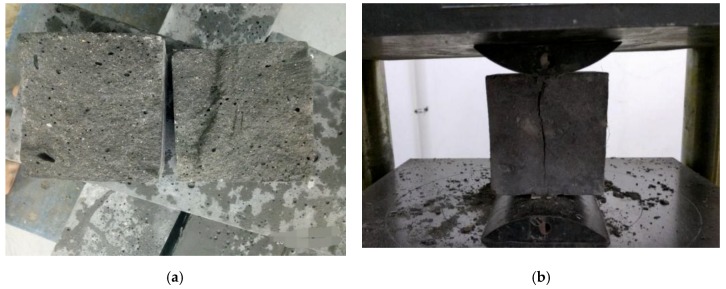
Splitting tensile failure morphology: (**a**) RPC, (**b**) carbon-fiber RPC.

**Figure 7 materials-12-03759-f007:**
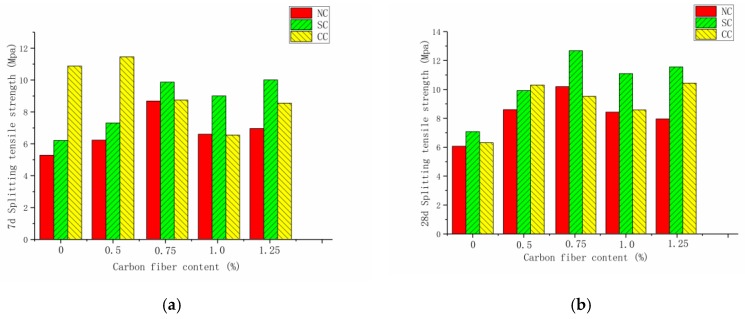
Splitting tensile strength under different curing systems at seven days and 28 days: (**a**) seven days, (**b**) 28 days.

**Figure 8 materials-12-03759-f008:**
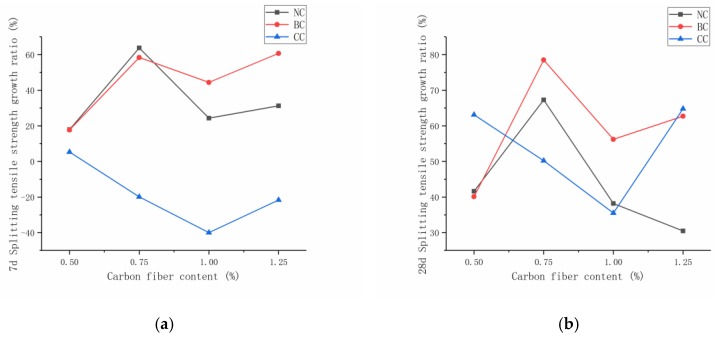
Splitting tensile strength growth ratio at seven days and 28 days: (**a**) seven days, (**b**) 28 days.

**Figure 9 materials-12-03759-f009:**
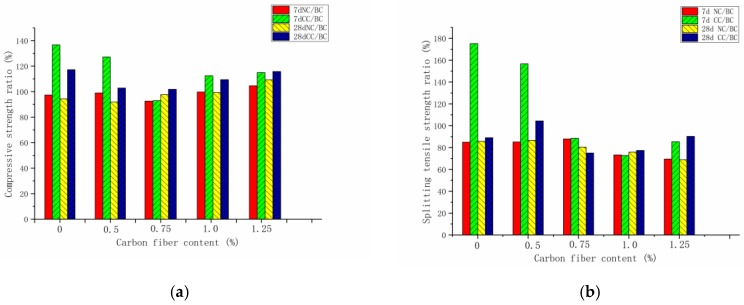
Carbon-fiber RPC strength versus standard curing strength ratio: (**a**) compressive strength ratio, (**b**) splitting tensile strength ratio.

**Figure 10 materials-12-03759-f010:**
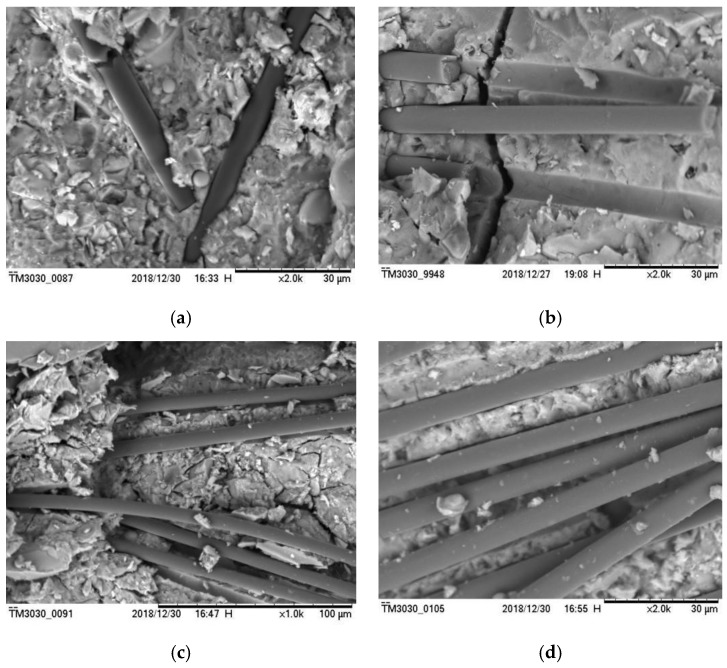
Picture of carbon fiber in cement base: (**a**) 0.5%, (**b**) 0.75%, (**c**) 1.0%, (**d**) 1.25%.

**Figure 11 materials-12-03759-f011:**
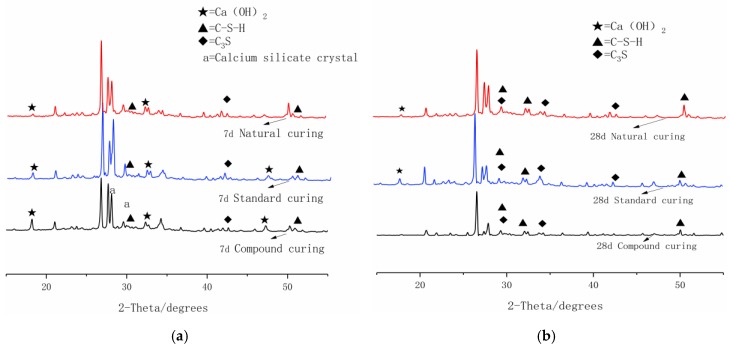
X-ray diffraction (XRD) analysis of three curing systems at seven days and 28 days: (**a**) seven days, (**b**) 28 days.

**Figure 12 materials-12-03759-f012:**
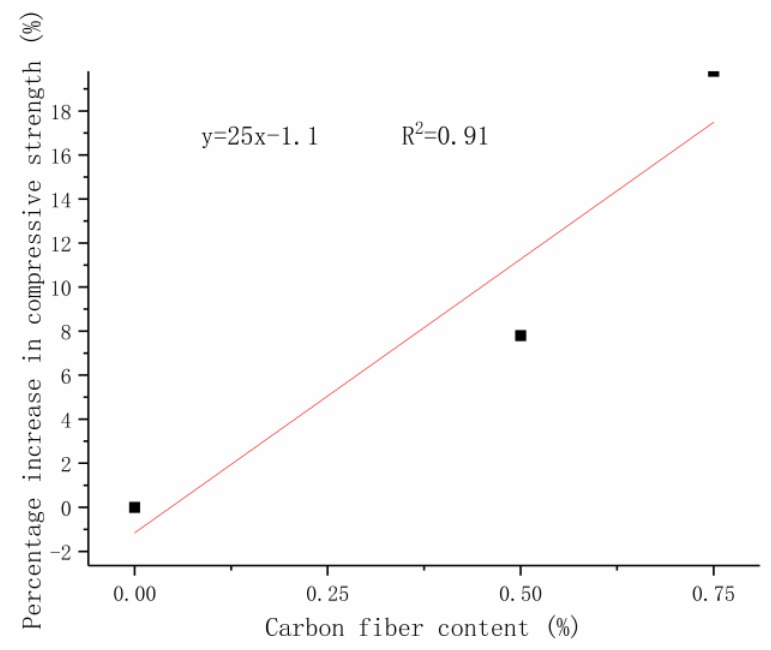
Relationship between compressive strength and fiber content.

**Figure 13 materials-12-03759-f013:**
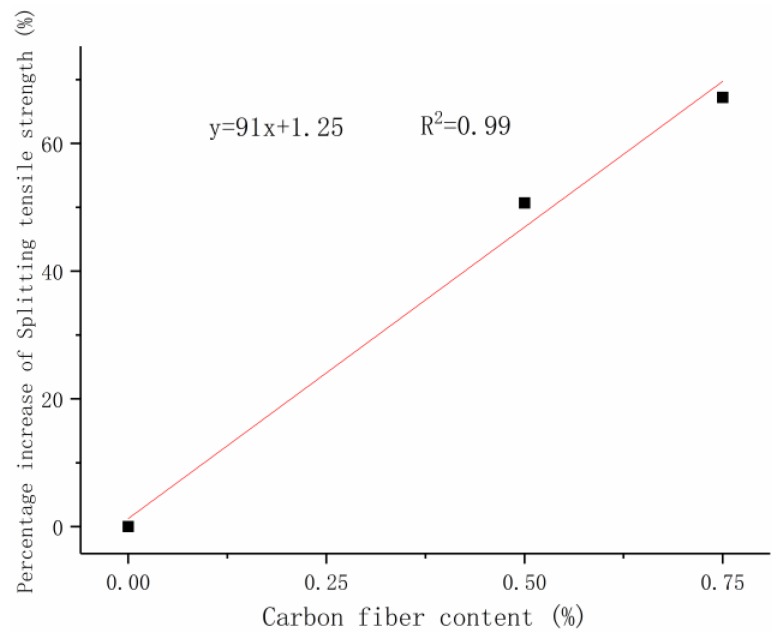
Relationship between splitting tensile strength and carbon-fiber content.

**Table 1 materials-12-03759-t001:** Chemical composition and physical properties of cement.

Name	Stability	Ignition Loss (%)	Specific Surface Area (m^2^/g)	MgO (%)	SO_3_ (%)	CaO (%)	Insolubles (%)	CI^−^ (%)
Cement	Qualified	1.61	367	0.98	2.62	61.7	1.01	0.007

**Table 2 materials-12-03759-t002:** Silica-fume index parameters.

Silica (%)	Ignition Loss (%)	CI^−^ (%)	Specific Surface Area (m^2^/g)	Moisture Content (%)	Water Demand Ratio (%)	Activity Index (%)
94.5	2.5	0.05	20	1.2	118	116

**Table 3 materials-12-03759-t003:** Slag-powder index parameters.

Name	Specific Surface Area (m^2^/g)	7-day Activity Index (%)	28-day Activity Index (%)	Density (g/cm^3^)	Ignition Loss (%)	Moisture Content (%)
Slag-powder	409	83	98	2.9	1.07	0.1

**Table 4 materials-12-03759-t004:** Index parameters of carbon fiber.

Density (g/cm^3^)	Tensile Strength (MPa)	Elongation (%)	Carbon Content (%)	Tensile Modulus of Elasticity (GPa)	Length (mm)
1.88	≥3000	1.5	93	≥200	7

**Table 5 materials-12-03759-t005:** Maintenance methods.

Maintenance Method		Specific Process
NC	Natural curing	Indoor natural curing was conducted from 1 day to 7 and 28 days.
SC	Standard curing	The specimens were maintained in the curing box for 7 and 28 days.
CC	Compound curing	The specimens were submerged in 90 °C hot water for 1 day and then transferred to the plant for natural curing for 7 and 28 days.

**Table 6 materials-12-03759-t006:** Mix proportions of carbon-fiber reactive powder concrete (RPC).

Curing Mode	Serial Number	Water/Binder Ratio	Cementing Material	Silica Fume (%)	Slag-Powder (%)	Fly Ash (%)	Water Reducer (%)	Sand/Cement Ratio	Carbon-Fiber Content (%)
NC	NCF-0	0.18	1	0.2	0.1	0.1	3	0.9	0
	NCF-1	0.18	1	0.2	0.1	0.1	3	0.9	0.5
	NCF-2	0.18	1	0.2	0.1	0.1	3	0.9	0.75
	NCF-3	0.18	1	0.2	0.1	0.1	3	0.9	1
	NCF-4	0.18	1	0.2	0.1	0.1	3	0.9	1.25
SC	SCF-0	0.18	1	0.2	0.1	0.1	3	0.9	0
	SCF-1	0.18	1	0.2	0.1	0.1	3	0.9	0.5
	SCF-2	0.18	1	0.2	0.1	0.1	3	0.9	0.75
	SCF-3	0.18	1	0.2	0.1	0.1	3	0.9	1
	SCF-4	0.18	1	0.2	0.1	0.1	3	0.9	1.25
CC	CCF-0	0.18	1	0.2	0.1	0.1	3	0.9	0
	CCF-1	0.18	1	0.2	0.1	0.1	3	0.9	0.5
	CCF-2	0.18	1	0.2	0.1	0.1	3	0.9	0.75
	CCF-3	0.18	1	0.2	0.1	0.1	3	0.9	1
	CCF-4	0.18	1	0.2	0.1	0.1	3	0.9	1.25

**Table 7 materials-12-03759-t007:** RPC working performance.

Test parameters						Workability
Water/Binder Ratio	Slag-Powder (%)	Silica Fume (%)	Fly Ash (%)	Sand/Cement Ratio	Carbon-Fiber Content (%)	Fluidity (mm)
0.18	10	20	10	0.9	0	235
					0.5	220
					0.75	210
					1.0	175
					1.25	155

**Table 8 materials-12-03759-t008:** Mechanical properties of carbon-fiber RPC.

Curing Mode	Serial Number	Cube Compressive Strength (MPa)		Cube Splitting Tensile Strength (MPa)	
		7 days	28 days	7 days	28 days
NC	NCF-0	71.80	77.49	5.28	6.07
	NCF-1	76.27	86.47	6.23	8.60
	NCF-2	80.86	99.24	8.65	10.16
	NCF-3	73.25	92.95	6.57	8.39
	NCF-4	74.34	92.16	6.93	7.92
SC	SCF-0	73.19	82.09	6.21	7.08
	SCF-1	77.07	91.98	7.31	9.92
	SCF-2	87.41	101.81	9.84	12.64
	SCF-3	73.48	93.74	8.97	11.06
	SCF-4	71.24	84.34	9.98	11.52
CC	CCF-0	100.06	96.26	10.88	6.31
	CCF-1	98.08	94.56	11.46	10.30
	CCF-2	81.21	104.73	8.71	9.48
	CCF-3	82.60	102.50	6.52	8.55
	CCF-4	81.90	97.64	8.52	10.40

**Table 9 materials-12-03759-t009:** Calculation of compressive strength of carbon-fiber RPC at 28 days.

Carbon-Fiber Content (%)	*λ**c* (%)	NC (MPa)	Error (%)	SC (MPa)	Error (%)	CC (MPa)	Error (%)
0	−1.1	76.63	−1.1	81.18	−1.1	95.20	−1.1
0.5	11.4	86.32	−0.17	91.48	−0.58	107.23	2.3
0.75	17.6	91.17	−7.9	96.59	−5.1	113.20	8.0

**Table 10 materials-12-03759-t010:** RPC compressive strength from References [27,28].

Carbon-Fiber Content (%)	Standard Curing			Hot Water Curing			Steam Curing		
	Actual Value (MPa)	Calculated Value (MPa)	Error (%)	Actual Value (MPa)	Calculated Value (MPa)	Error (%)	Actual Value (MPa)	Calculated Value (MPa)	Error (%)
K0	94.8	93.8	−1.1	122	120.66	−1.1	145.35	143.76	−1.1
K0.5	111.2	105.6	−5.0	130	135.9	4.5	-	-	-
G0	130.1	128.6	−1.1	133	131.5	−1.1	142.0	140.4	−1.1
G0.5	136.6	144.9	6.1	140	148.1	5.5	150.0	158.2	5.48.

K: Ke; G: Guo.

**Table 11 materials-12-03759-t011:** Calculated splitting tensile strength of carbon-fiber RPC at 28 days.

Carbon-Fiber Content (%)	*λ**_t_* (%)	NC (MPa)	Error (%)	SC (MPa)	Error (%)	CC (MPa)	Error (%)
0	1.25	6.14	1.3	7.17	1.25	6.38	1.25
0.5	46.8	8.9	3.5	10.4	4.9	9.38	−9.5
0.75	69.5	10.28	1.2	12.64	−5.1	10.30	−7.8

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
