# Peer review of "Mechanical Properties of Carbon-Fiber RPC and Design Method of Carbon-Fiber Content under Different Curing Systems"

_materials, 2019, doi:10.3390/ma12223759_

Round 1

Reviewer 1 Report

The paper "Mechanical Properties of Carbon Fiber RPC and Design Method of Carbon Fiber Content under Different Curing Systems" is interesting and within the scope of Materials.

I have some questions for the authors:

please improve notations of units in tables; please use apex and subscript to declare correctly units of measure; the text is not compliant with the template of the journal (e.g. lines 119-121); Figure 1 should be improved; please explain the mean of a), b), c) in the caption of Figure 2; values of bars in Figure 9 are not readable.

Author Response

1.Please improve notations of units in tables; Reply:Modified as required 2.Please use apex and subscript to declare correctly units of measure; Reply:Modified as required 3.The text is not compliant with the template of the journal (e.g. lines 119-121); Reply:Modified as required 4.Figure 1 should be improved; Reply:Modified as required 5.Please explain the mean of a), b), c) in the caption of Figure 2; Reply:Modified as required 6.Values of bars in Figure 9 are not readable.  Reply:Modified as required Thank you,professor!

Reviewer 2 Report

My main concern about the paper is that the authors used a water to binder ratio of 0.18 and for such a lower ratio and without using any coarse aggregates in manufacturing the concrete, it is reasonable to expect a 28 day cylindrical compressive strength of 150 MPa even under normal curing without heat and pressure.  Table 6 shows that the maximum 28d compressive strength using cubes was about 105 MPa ( may be only 90 MPa using cylinders).  Therefore the authors should explain the reasons for this as otherwise using such a high amount of cement and specialist curing techniques is a huge disadvantage for producing only a high strength concrete and not a very high strength concrete.

Author Response

1.My main concern about the paper is that the authors used a water to binder ratio of 0.18 and for such a lower ratio and without using any coarse aggregates in manufacturing the concrete, it is reasonable to expect a 28 day cylindrical compressive strength of 150 MPa even under normal curing without heat and pressure.  Table 6 shows that the maximum 28d compressive strength using cubes was about 105 MPa ( may be only 90 MPa using cylinders).  Therefore the authors should explain the reasons for this as otherwise using such a high amount of cement and specialist curing techniques is a huge disadvantage for producing only a high strength concrete and not a very high strength concrete.

Reply:For convenience of promotion and to meet the actual engineering needs, the relatively common and cheap class II fly ash and slag powder are used in this test.Under the standard curing condition, the compressive strength of the cube can reach 101.81Mpa; under the composite curing condition, the compressive strength of the cube can reach 105MPa, which  meets the production requirements of RPC100 in GB/T 31387-2015.

Thank you,professor!

Reviewer 3 Report

This paper investigates the effect of different curing systems on the mechanical properties and microstructure of carbon fibre reinforced reactive powder concrete. The topic is suitable for Materials. The state-of-the-art part is acceptable as the most important references are included. More detailed explanation and discussion on the test results are required to enhance the readability and understanding of this study. In addition, a native English speaker should correct the manuscript before submitting. Detailed comments are listed further down:

Page 1 Line 36,39 etc. – Please do not use the semicolons. Please explain the workability of the RPCs. Page 5 – Please think about it and improve: “Numerous tests on the basic mechanical properties, including cubic compressive and tensile strength tests, under different curing systems on the 7th and 28th days of carbon RPC.” Please correct the unit in Table 6. Page 6 – Please explain more clearly “… the concrete continuously drops the slag during the load increase, … cracks are gradually generated, and development starts from both sides.” Please improve the titles of the Figures. Figures 5 and 8 – Please explain, why were the results connected by curved lines? Figures 4 and 7 – Please consider again and improve “Percentage increase”. Please consider presenting these results more clearly. Page 7 – Please consider whether in any case: “the optimal blending amount is achieved when the carbon fiber content is 0.75%.” Page 7 – Please check whether the statement is correct or not: “The increase under standard curing conditions is −2.6% to 19.4%.” Page 7 – Please also consider the effect of carbon fibre orientation: “When the carbon fiber content is greater than the critical value of 0.75%, the fiber agglomeration inside the RPC matrix is severe, thereby resulting in an increase in the internal defects and a decrease in compressive strength.” Please give relevant references. Page 8 – Please think about CC curing mode: “the Figure 8 illustrates that the tensile strength increases with the carbon fiber content when the carbon fiber content is 0%–0.75%. The compressive and splitting tensile strengths of 7 and 28 d gradually increase.” Page 8 – Please think about NC curing mode: “The strength rises at 1% to 1.25%.” Page 9 – Please improve: “The ratio of strength.” Page 9 – Please consider again and improve: “The fiber content of 0.75% is a critical value. The compressive and tensile strengths accounted for 93% to 136% and 90% to 168% of the standard curing, respectively, before 0.75%.” Page 12 – Please explain on what basis it was found that: “The carbon fiber has good compatibility with the cement, and the bonding is relatively tight.” Page 13 – Please consider again and improve the Conclusion (1). The quality of English requires further improvement. Please improve the References according to journal requirements.

Author Response

1.Page 1 Line 36,39 etc. – Please do not use the semicolons.

Reply:Modified as required

2.Please explain the workability of the RPCs.

Reply:Modified as required

3.Page 5 – Please think about it and improve: “Numerous tests on the basic mechanical properties,

Reply:Modified as required

4.including cubic compressive and tensile strength tests, under different curing systems on the 7th and 28th days of carbon RPC.”

Reply:Modified as required

5.Please correct the unit in Table 6. Page 6 – Please explain more clearly “… the concrete continuously drops the slag during the load increase, … cracks are gradually generated, and development starts from both sides.”

Reply:Modified as required

6.Please improve the titles of the Figures.

Reply:Modified as required

7.Figures 5 and 8 – Please explain, why were the results connected by curved lines?

Reply:Modified as required

It's been changed back to a line chart,I thought it was a little smoother.After listening to your advice,I think that the line chart is more intuitive and clear.

Figures 4 and 7 – Please consider again and improve “Percentage increase”.

Reply:Modified as required

9.Please consider presenting these results more clearly.

Reply:Modified as required

10.Page 7 – Please consider whether in any case: “the optimal blending amount is achieved when the carbon fiber content is 0.75%.”

Reply:Modified as required,This conclusion has been corrected.

11.Page 7 – Please check whether the statement is correct or not: “The increase under standard curing conditions is −2.6% to 19.4%.”

Reply:Modified as required,This conclusion has been corrected.

12.Page 7 – Please also consider the effect of carbon fibre orientation: “When the carbon fiber content is greater than the critical value of 0.75%, the fiber agglomeration inside the RPC matrix is severe, thereby resulting in an increase in the internal defects and a decrease in compressive strength.” Please give relevant references.

Reply:Modified as required

Combined with the microscopic analysis, the explanation is given in the microscopic analysis of the fourth section

13.Page 8 – Please think about CC curing mode: “the Figure 8 illustrates that the tensile strength increases with the carbon fiber content when the carbon fiber content is 0%–0.75%.

Reply:Modified as required

14.The compressive and splitting tensile strengths of 7 and 28 d gradually increase.” Page 8 – Please think about NC curing mode: “The strength rises at 1% to 1.25%.”

Reply:Modified as required

15.Page 9 – Please improve: “The ratio of strength.”

Reply:Modified as required

16.Page 9 – Please consider again and improve: “The fiber content of 0.75% is a critical value. The compressive and tensile strengths accounted for 93% to 136% and 90% to 168% of the standard curing, respectively, before 0.75%.”

 Reply:Modified as required

17.Page 12 – Please explain on what basis it was found that: “The carbon fiber has good compatibility with the cement, and the bonding is relatively tight.”

Reply:Modified as required   

The explanation is given in the microscopic analysis of the fourth section

18.Page 13 – Please consider again and improve the Conclusion (1). The quality of English requires further improvement. Please improve the References according to journal requirements. 

Reply:Modified as required

Thank you very much. Your guidance makes me very important to me. Thank you again, professor

Reviewer 4 Report

The article needs major grammatical and syntax improvements. Use of English service center is recommended. A few examples of the English errors are as follows (These are just a few examples and formatting has some issues)

“cement mortar containing coarse aggregate and high water-to-binder ratio.” Should be the cement mortar and tensile strengths meet actual engineering requirements and provides the calculation It is suggested that the introduction would have better flow in terms of works that are related to the current study, rather than mentioning the names and brief explanations of what has been done in a specific work The introduction is limited, at least a couple more paragraphs are needed for further discussion. It is suggested to just briefly mention about the other environmental effects and expand the introduction on the behavior of the concrete. A couple of references that is suggested for inclusion are as follow: Farzampour, Alireza. "Compressive Behavior of Concrete under Environmental Effects." IntechOpen, 2019 Farzampour, Alireza. "Temperature and humidity effects on behavior of grouts." Advances in concrete construction, An international journal 5.6 (2017): 659-669. The material properties are based on which code? Why these specified properties are used for investigation? Water cement ratio plays an important role in the overall strength of the specimens. Why 0.18 is used for further investigations? What will happen if the mix design is made with lower water binder ratios How many specimens are considered to capture the strength at each fiber content? Comparing Fig5 c and d, with Fig 8 c and d, ( especially for the CC trends) it is highly suggested that the validity of the data to be confirmed once more and if the number of specimens are not enough, more investigations should be considered to confirm these results. Fig 4 is vague, it is suggested that to use similar 2D figure (similar to figure 5) and instead use higher number of figures to clarify the subject. Figure 5 legend box is incomplete. What is the reason that 0.75 CF% has the best performance in achieving more strength compared to the rest of the fiber percentages? what is the difference in the physical behavior? From Figure 4, by increasing CF content, why CC follows decreasing trend and NC and BC follow increasing trends? Fig 7 could be changed, it is suggested that to use similar 2D figure (similar to fig 5) and instead use higher number of figures to clarify the subject. Did authors conduct any maturity test evaluation to investigate the strength development procedure under different curing procedures? For this type of investigation, the maturity investigation is highly suggested. In Fig 8d, the CC model would follow decreasing trend compared to NC and BC by increasing the CF content. What is the reason for that? The R2 values reported in Fig 10 seems to be low, why? And is there way to improve that? What does “internal defects gradually occur” mean? Description of the XRD should be added and why it is conducted should be elaborated. It is highly suggested that the authors consider observation results for Fig 5 and Fig 8 in SEM microstructural analysis. How would the author justify the Fig 5 and Fig 8 based on the SEM results? For the XRD Analysis, the y axis is intensity which should be added to the figures. More quantitative results should be mentioned rather than qualitative results in the conclusion

Author Response

“cement mortar containing coarse aggregate and high water-to-binder ratio.” Should be the cement mortar and tensile strengths meet actual engineering requirements and provides the calculation It is suggested that the introduction would have better flow in terms of works that are related to the current study, rather than mentioning the names and brief explanations of what has been done in a specific work The introduction is limited, at least a couple more paragraphs are needed for further discussion.

Reply:Previous research focused on carbon fiber concrete (containing coarse aggregate), high water-binder ratio of carbon fiber cement mortar.However, the mechanical properties of both of them are lower than that of carbon fiber RPC, and the research on carbon fiber reinforced RPC under different curing methods is less and imperfect.

It is suggested to just briefly mention about the other environmental effects and expand the introduction on the behavior of the concrete. A couple of references that is suggested for inclusion are as follow: Farzampour, Alireza. "Compressive Behavior of Concrete under Environmental Effects." IntechOpen, 2019 Farzampour, Alireza. "Temperature and humidity effects on behavior of grouts." Advances in concrete construction, An international journal 5.6 (2017): 659-669.

Reply:Modified as required,Already referenced

3.The material properties are based on which code?

Reply:GB/T 31387-2015

Why these specified properties are used for investigation? Water cement ratio plays an important role in the overall strength of the specimens. Why 0.18 is used for further investigations?

Reply:As you said, the water-binder ratio is an important factor.Before this test, our team did a lot of test mix ratio tests, and these factors (water-binder ratio, sand-binder ratio, silica fume, slag powder) were based on the previous test. According to the comprehensive compressive strength, fluidity and working performance, the water-binder ratio was set at 0.18.

What will happen if the mix design is made with lower water binder ratios ?

Reply:Too low water-binder ratio (< 0.18) will lead to difficulties in stirring, increasing difficulty in forming test blocks and increasing discrete type.The water - binder ratio of 0.18, with 3% efficient superplasticizer, working performance meet the requirements.

How many specimens are considered to capture the strength at each fiber content?

Reply:In each curing system, 12 specimens with fiber content were prepared, and the compressive strength and splitting strength were tested at 7d and 28d respectively.The three test blocks are a group and are averaged (error specimens are deleted).

6.Comparing Fig5 c and d, with Fig 8 c and d, ( especially for the CC trends) it is highly suggested that the validity of the data to be confirmed once more and if the number of specimens are not enough, more investigations should be considered to confirm these results.

Reply:From preparation to test, we spent nearly a year.This test data is real, and the experimental amount is large enough, because we have a large team, and all the experiments are done by ourselves.We in good faith, pragmatic principles to explore the test.

Fig 4 is vague, it is suggested that to use similar 2D figure (similar to figure 5) and instead use higher number of figures to clarify the subject.

Reply:Modified as required.

8.Figure 5 legend box is incomplete.

Reply:Modified as required

What is the reason that 0.75 CF% has the best performance in achieving more strength compared to the rest of the fiber percentages? what is the difference in the physical behavior?

Reply:It is explained in the microscopic analysis.With 0.75% fiber content, carbon fiber is evenly dispersed, and the fiber and cement matrix bond closely. While more fibers are under external load, more fibers are pulled and absorb more energy.When the fiber content exceeds 0.75%, the fibers begin to overlap and cluster, which is not conducive to the development of strength.

From Figure 4, by increasing CF content, why CC follows decreasing trend and NC and BC follow increasing trends?

Reply:Dear professor, yes. First, the composite curing in this paper is 1d hot water curing +27d natural curing.When the transition from hot water curing to natural curing occurs, the temperature difference leads to a high temperature and high pressure environment for the capillary pores, significantly lower bound water content, and RPC is rough and porous. Therefore, the early incorporation of fibers could not make up for the defects.With the progress of hydration, the strength returned to normal at 28d.Both natural curing and standard curing conditions are normal strength improvement.

11.Fig 7 could be changed, it is suggested that to use similar 2D figure (similar to fig 5) and instead use higher number of figures to clarify the subject.

Reply:Modified as required.

Did authors conduct any maturity test evaluation to investigate the strength development procedure under different curing procedures? For this type of investigation, the maturity investigation is highly suggested.

Reply:Due to the limited laboratory equipment, there is no mature testing equipment.I will listen to your advice carefully.

In Fig 8d, the CC model would follow decreasing trend compared to NC and BC by increasing the CF content. What is the reason for that? 

Reply:From 1d90℃ hot water to natural curing, considering from the actual project, the cooling speed is accelerated to meet the requirements of the accelerated construction period of the actual project.This is because the temperature difference effect in the capillary pores formed high temperature and high pressure environment.The capillary interstice increases, the bound water decreases obviously, the fiber content increases unceasingly, the internal defect increases, the strength decreases.At 28d, with the progress of hydration, the compactness increased and the strength returned to normal.

14.The R2 values reported in Fig 10 seems to be low, why? And is there way to improve that?

Reply:In order to facilitate the acceptance of front-line technicians, straight line fitting is adopted to increase the readability of the formula and facilitate the popularization of the project.R2=0.91 in this paper, which also has a certain correlation. R2 value was obtained by fitting the experimental data in this paper.

What does “internal defects gradually occur” mean?

 Reply:Modified as required.It means that internal defects begin to occur.

16.Description of the XRD should be added and why it is conducted should be elaborated.

Reply:Modified as required.

It is highly suggested that the authors consider observation results for Fig 5 and Fig 8 in SEM microstructural analysis.

Reply:Modified as required.It has been explained in SEM chapter.

How would the author justify the Fig 5 and Fig 8 based on the SEM results?

Reply:It has been explained in SEM chapter.

19.For the XRD Analysis, the y axis is intensity which should be added to the figures. More quantitative results should be mentioned rather than qualitative results in the conclusion

Thank you,professor!

Round 2

Reviewer 1 Report

The paper can be accepted

Reviewer 3 Report

Authors have reasonably improved the manuscript as per suggestions of reviewer. Now, I could recommend the paper for publication after minor editorial corrections required.

Reviewer 4 Report

NA